# Vibration Monitoring of the Mechanical Harvesting of Citrus to Improve Fruit Detachment Efficiency

**DOI:** 10.3390/s19081760

**Published:** 2019-04-12

**Authors:** Sergio Castro-Garcia, Fernando Aragon-Rodriguez, Rafael R. Sola-Guirado, Antonio J. Serrano, Emilio Soria-Olivas, Jesús A. Gil-Ribes

**Affiliations:** 1Department of Rural Engineering, Universidad de Cordoba, 14004 Cordoba, Spain; g92arrof@uco.es (F.A.-R.); ir2sogur@uco.es (R.R.S.-G.); gilribes@uco.es (J.A.G.-R.); 2IDAL, Intelligent Data Analysis Laboratory, Universidad de Valencia, 46100 Valencia, Spain; ajserran@uv.es (A.J.S.); emilio.soria@uv.es (E.S.-O.)

**Keywords:** *Citrus sinensis* L. Osbeck, mechanical harvesting, acceleration sensor, vibration time, logistic regression

## Abstract

The introduction of a mechanical harvesting process for oranges can contribute to enhancing farm profitability and reducing labour dependency. The objective of this work is to determine the spread of the vibration in citrus tree canopies to establish recommendations to reach high values of fruit detachment efficiency and eliminate the need for subsequent hand-harvesting processes. Field tests were carried out with a lateral tractor-drawn canopy shaker on four commercial plots of sweet oranges. Canopy vibration during the harvesting process was measured with a set of triaxial accelerometer sensors with a datalogger placed on 90 bearing branches. Monitoring of the vibration process, fruit production, and branch properties were analysed. The improvement of fruit detachment efficiency was possible if both the hedge tree and the machinery were mutually adjusted. The hedge should be trained to facilitate access of the rods and to encourage external fructification since the internal canopy branches showed 43% of the acceleration vibration level of the external branches. The machine should be adjusted to vibrate the branches at a vibration time of at least 5.8 s, after the interaction of the rod with the branch, together with a root mean square acceleration value of 23.9 m/s^2^ to a complete process of fruit detachment.

## 1. Introduction

Citrus fruits, whether for fresh consumption or industrial processing, are mainly harvested by hand. Worldwide, 147 million tonnes of citrus were produced in 2017, including orange, grapefruit, lemon, mandarin, and other citrus fruits [1]. Spain is the sixth largest producer of citrus fruit in the world, with an approximate production in 2017–2018 of more than seven million tonnes. In Spain, the predominant citrus orchards are trained for manual harvesting, with an orientation towards the fresh market. Manually harvested orchards experience problems due to the availability of labour and the high cost of operation.

Within the citrus production process, harvesting is a phase of enormous economic importance due to its high impact on the final cost of production. The manual harvesting in Southern Spain requires an average of 95 days’ work per hectare and represents between 25% and 35% of the final cost of production [2]. Roka and Hyman [3] stated that, under Florida conditions, the application of mechanical harvesting for industrial processing could provide a 50% cost reduction, while increasing labour productivity by ten per cent. With the current approach to citrus production, the high costs of manual harvesting could compromise the profitability of the activity and the future of plantations in the long term [4].

Since the 1970s, the development of mechanical citrus harvesting systems for the juice industry has mainly taken place in Florida. However, none of the mechanised systems have been able to match the flexibility and fruit selection capabilities of manual harvesting [5]. The foremost mechanical harvesting systems are the trunk and canopy shaker systems that were applied to and developed for citrus fruits, and which reach high values of harvesting efficiency ranging between 84–95% and 55–95%, respectively [6]. However, the detachment of immature fruitlets has been identified in both systems; this occurs particularly in the late varieties that are of special interest to the juice industry and, in addition to a reduction in the working capacity (ha/h), it could represent an obstacle to the adoption of these mechanical harvesting systems by farmers. Roka et al. [7] showed yield reduction values for the use of these machines compared to manual harvesting of 20–50% according to their use and regulation. In parallel, the development and testing of abscission agents have permitted an increase in the working capacity and an improvement in the detachment of mature fruit with these harvesting technologies [8]. It was shown that a moderate reduction in fruit detachment force, through the application of an abscission agent, was enough to significantly increase the harvesting efficiency [9]. Subsequently, it was demonstrated that the use of an abscission agent together with an adjustment of the vibration parameters, fundamentally time and frequency of vibration, allowed high percentages of mature fruit detachment to be achieved without significantly affecting the following season [10]. This result was confirmed based on the different frequency responses of mature fruit and immature fruitlets to mechanical harvesting [11].

Canopy shaker systems allow a continuous vibration of the tree row; the rods penetrate the canopy and achieve a high value of fruit detached in areas where there is direct contact of rods with branches. The use of a canopy shaker can generate a greater fall of leaves, shoots, and branches than manual harvesting. The fall of these organs is considered as tree damage, which could have negative implications in the yield, the productive life, and the cost of transport from the orchard to the industry [12]. For this reason, the improvement of these machines has been based on a dual objective; to increase the efficiency of mature fruit removal and to reduce the damage caused to trees. In order to improve the mechanical harvesting process, Savary et al. [13] developed a canopy shaker simulation based on finite element methods to predict and evaluate the interaction between the tree and the machine. Then, Savary et al. [14] evaluated the effect of vibration on the tree canopy according to the distribution of forces and accelerations in branches and fruits. Subsequently, machine improvement proposals were based on mathematical models and prototype tests with a combination of machine operating parameters such as frequency and amplitude of vibration [15,16], and the configuration and material of the rods [17]. The rods have been shown to play an essential role in the process of tree shaking, both in terms of fruit detachment and the possible generation of tree and fruit damage. Liu et al. [17] indicated that rod material affected the vibratory response of trees with respect to the acceleration peaks in branches. They recommended that the rods have high stiffness values, but that their surface should be smooth to reduce damage. The shape of the rods revealed that arc-shaped flexible rods had better performance with respect to fruit detachment efficiency and a lower rate of damage to trees than the free end rods [18]. In a further attempt to adapt the vibration process to tree requirements, Pu et al. [18] designed and tested a two-section independent shaker system in order to minimise tree damage and maximise harvesting efficiency. The adaptation of the machine to the variability of the tree canopy made it possible to reduce tree damage compared to other canopy shaker systems.

During the harvesting process, the machine rods penetrate the tree but do not usually reach all parts of the canopy, so vibration must be transmitted via the branches in order to detach the inner fruit. Whitney et al. [19] indicated that a greater fruit detachment efficiency value was obtained by harvesting smaller rather than larger canopy trees. Canopy shaker systems generally have a limited capacity to detach internal fruits from the tree and yet can remove almost all external fruits. The harvesting process of fruit inside the canopy may cause damage to the outermost parts due to a reduction in machine ground speed or an increment in vibration frequency value and may also require additional manual harvesting.

This study works on the hypothesis that mechanised harvesting processes can be carried out with a high value of harvesting efficiency, where subsequent hand-harvesting is not necessary, and that the possible, but moderate tree damage does not affect the productive life of the orchard. The objective of this work is to analyse the interaction of the canopy shaker rods with the tree branches from the data collected continuously during the mechanical harvesting process of citrus trees. First, the fruit-bearing branches with and without direct contact with the machine rods were characterised in the position and production. Then, the canopy vibration process during mechanical harvesting was monitored. Finally, the vibration parameters necessary to detach the fruit were determined, and recommendations were given on the harvest parameters necessary to increase the fruit removal efficiency.

## 2. Materials and Methods

Mechanical harvest tests were carried out in Cordoba (Spain) in 2017 during the sweet orange (*Citrus sinensis* L. Osbeck cv. Valencia) harvest season for juice production, during four weeks from the end of flowering to before the natural fall of immature fruitlets in June. Table 1 shows the main characteristics of the four, mechanically harvested citrus orchards. Trees had been planted in wide hedges over 0.4 m ridges and had wide row distances to allow machine manoeuvrability and the use of canopy shaker harvesters (Figure 1).

Mechanical harvesting was carried out with a lateral tractor-drawn continuous canopy shaker system (Oxbo 3210, Byron, New York, NY, USA), working under regular conditions, with a ground speed range between 1 and 1.5 km/h (0.28–0.42 m/s), and a vibration frequency close to 4.5 Hz, which caused the fruit to fall to the ground. Harvesting tests were carried out to ensure close contact of the shaker system with the tree canopy (Figure 2). The machine harvested both sides of the hedge in independent passes, with an approximate working capacity of 0.4–0.5 ha/h. Subsequently, the fruit remaining in the canopy was hand-picked and collected together with the fruit from the ground and loaded into a container.

Interaction of the harvesting system with the tree canopy was studied for each side of the tree hedges (Figure 3). Because the tree hedges differed between orchards, a representative cross-section of the tree canopy was selected. The cross-section ranged from 4–5 m^2^, and was composed of a distance from the line of the trunks to the outside of the canopy of up to 2.0–2.5 m, and of a height ranging from 0.5–2.5 m. This cross-section was selected because it differentiated an external area of the canopy with direct contact of the machine rods with the branches, for a rod length of 1.4 m, and an internal area of the canopy without direct contact with the rods. Furthermore, the cross-section was representative of the tree canopy, with high yield, avoiding the effect of the lower pendulous branches and allowing analysis of the vibration process in a homogeneous canopy area between the tested plots. The cross-section was divided into 16–20 sectors according to the width of the hedge, at intervals of 0.5 m both horizontally and vertically. In each sector, the values of vibration, the properties of the branches, and the number of fruits were recorded. The canopy area that had direct contact with the rods was sampled with 10 sectors, while the canopy area without direct contact was sampled with 6 or 10 sectors.

Before the mechanical harvesting process, a total of 90 fruit-bearing branches were selected that had mature fruit and were distributed in different sectors. The statistical design established a stratified random sampling, each cluster was a tested plot, and in each plot, 18–24 fruit-bearing branches were randomly selected. The sample guaranteed at least three measurements in each sector. Each branch was assigned a position value in the cross-section at a point close to the fruit that was able to support an acceleration sensor, but which had a diameter less than 10 mm. The fruit detachment ratio was determined by the number of fruits removed from each branch before and after the harvesting process.

Branch vibration measurements were recorded with a triaxial MEMS accelerometer sensor (Gulf Coast Data Concepts LLC X200-4, Waveland, MS, USA) with a measurement range of ±2000 m/s^2^, a 16-bit resolution, a sensitivity of 0.06 m/s^2^, and a sampling frequency of 400 Hz. Figure 4 shows the placement of the sensor on the fruit-bearing branch.

Analysis of the acceleration signals in the time domain and statistical analysis was performed using the R open software (R Core Team, 2016) and in the frequency domain using the NV Gate v8.0 software, with a fast Fourier transformation with 401 lines in a frequency range of 0–156.2 Hz. In the time domain, the resultant acceleration value (A_r_) was determined as the module of the vector sum of the three measurement axes in each sensor. Figure 5 shows a sample of A_r_ in a fruit-bearing branch.

In the time domain, the vibration variables studied were:
■Vibration time (T_vib_): time (s) elapsing between the first and last value of A_r_ measured on the branch, ranging from A_r_ values of 20 m/s^2^ up to 600 m/s^2^.■Mean peak acceleration (A_pk_): the mean value of the 10 maximum peak values of A_r_ (m/s^2^) for T_vib_20_.

In the frequency domain, the vibration variables studied were:
■Frequency: number of cycles per second (Hz) of rod movement in the canopy.■RMS acceleration (A_RMS_): vector sum of the Root Mean Square values (RMS) of each accelerometer axis at the vibration frequency.

The statistical analysis focused on predicting the fruit detachment ratio according to the vibration variables and the branch measurements as the predicted parameters. Logistic regression was used with a K = 2-fold cross-validation method.

## 3. Results and Discussion

Most of the fruit (72.7%) was located in the canopy area that had direct contact with the rods, in the height range of 1–2 m from the ground, and in a range of 0.5–2 m from the trunk. Gupta et al. (2015) stated that the area with the highest fructification is in the primary branches of the intermediate zone of the canopy at a height of 1.14–2.29 m, and at a distance of 0.78–0.83 m from the canopy exterior. However, the distribution of fruit in the canopy also depends on planting distances and tree height. A reduced distance between trees can generate a greater percentage of fruit in the upper canopy parts but can reduce the number of fruits inside [19].

The diameter of the branch at the vibration measurement point was 7.9 ± 2.4 mm (mean ± sd), with a variation that ranged from 10.2 mm for the branches closest to the trunk and the ground to 5.53 mm for the outermost and tallest branches of the canopy. Each tested branch carried a mean value of 3.7 ± 1.8 fruits. The results showed a high variability in the distribution of fruit in the canopy and in the morphology of the branches. This variability is important for the outcome of mechanical harvesting systems and was considered by Gupta et al. [15,16] for modelling the tree and simulating the harvesting process in order to improve the canopy shaker system. However, the current harvesting systems based on canopy shakers do not contemplate the variability of branches and fruit within the tree canopy. In an attempt to improve the machine adaptation to the tree, Pu et al. [18] designed and tested a canopy shaker system capable of applying different vibration parameters to the upper and lower parts of the tree. These authors showed the need to use different harvesting parameters and were able to achieve a high fruit detachment ratio (82.6%) with low tree damage.

Canopy shaker systems continuously harvest fruit as they move along the row of trees. Before the machine comes into contact with a branch, the branch may vibrate due to contact with other branches or due to the transmission of vibration from the trunk. Then, the branch vibrates due to contact with the machine rods, and finally, the branch vibrates freely when the machine has passed. Table 2 shows the results of this vibration process measured in branches, both in the canopy area with and without direct contact with the machine rods. In order to define the beginning and end of the vibration process, the acceleration values produced only by natural sources, mainly by wind and gravity, and without machine interaction were recorded. The vibration time where the branch was excited by the machine was defined as the time elapsing between resultant acceleration values greater than 18 m/s^2^ (T_vib_18_). The average vibration time (T_vib_18_) of the branches was 14.3 ± 2.8 s. No significant differences were found (Student’s *t*, *p* > 0.05) between the vibration time of the branches located in the canopy area with or without direct contact with rods. This indicated that all branches vibrated at the same time, but not all at the same level of acceleration. The length of the machine’s shaking system, i.e., the length of the rods and drums, together with the machine ground speed, determined the vibration time during which the rods had direct contact with the branches, whereas tree-training and canopy density could define the vibration time during which the branch vibrated without direct contact with the rods. For the field test conditions, i.e., a rod length of 1.4 m and ground speed ranging from 1–1.5 km/h (0.28–0.42 m/s), the canopy shaker systems were able to maintain direct contact with the branches within the range of 7.7–11.5 s. This indicated that between 20% and 46% of the vibration time may correspond to the transmission of vibration before and after the passage of the machine.

During the harvesting process, the branches in contact with the rods experienced a process of forced vibration. The rods, powered with an alternating and rotating movement, penetrated the canopy, producing an impulse excitation in the branches of the fruit of the outermost part of the canopy [20]. The branches showed a mean vibration frequency value of 4.1 ± 0.5 Hz. The value of the vibration frequency did not correlate with the position of the branch in the canopy (Pearson = 0.135, *p* > 0.05). The frequency value used was within the range recommended for citrus detachment with canopy shakers. Liu et al. [17] established that the value of 5 Hz was appropriate for the detachment of fruit without increasing the damage caused to the tree. Similarly, it was demonstrated that a frequency value of 4.8 Hz was adequate to produce a high acceleration in the branches when using rigid rods [18]. The frequency of the vibration in combination with the design of the rods used and the state of the immature fruits [10] play an important role in the damage caused to the trees. Although tree damage is an important element in the process of improving mechanised harvesting, it has not been considered in this study because the damage that occurred was minor and similar to that of previous years, mechanically harvested plantations had not shown any problems in tree development or yield compared to previous seasons.

The values of A_RMS_ measured in the branches was positively related to the fruit detachment ratio. The branch vibration was characterised by an A_RMS_ value of 26.5 ± 13.6 m/s^2^ for the machine vibration frequency. However, there was a significant variation in the mean A_RMS_ values in branches depending on their position in the canopy (Figure 6). Branches with direct contact with rods showed a significantly higher mean A_RMS_ value (29.6 ± 10.2 m/s^2^) (Student *t*, *p* < 0.05; Wilcoxon–Mann–Whitney, *p* < 0.05) than branches without direct contact (12.8 ± 6.4 m/s^2^). Pu et al. (2018) showed that the highest values of acceleration in the branches (31.4 m/s^2^) were provided by contact with the machine rods and these branches reached the highest values of fruit removal efficiency.

During the harvesting process, the acceleration produced in the branch was transmitted to the fruits for their detachment. The internal branches, without direct contact with the rods, showed 43% of the A_RMS_ vibration level of the external branches. A similar result was achieved by Liu et al. [17], whose results showed a reduction of the acceleration in the inner branches of the canopy of 42% with respect to the outer branches. In field trials with measurements inside the fruit before its detachment with canopy shaker systems, Castro–Garcia et al. [20] recorded mean A_RMS_ values between 38.8 and 51.3 m/s^2^. These values indicated that there was an amplification of the acceleration values from the branch to the fruit. Savary et al. [13] evaluated the acceleration produced in the branches and indicated that the resultant acceleration values were higher at the ends of the branches, especially on the thinnest and outermost branches. These same authors pointed out that the trunk region of unbranched trees had very low acceleration values, whereas the acceleration values began to be noticeably higher from the first branch of the trunk region.

The interaction of the rods with the branches was characterizsed by a succession of impacts with a high acceleration value in accordance with the vibration frequency of the machine. These impacts presented a mean A_pk_ value of 495.1 ± 270.9 m/s^2^. Similarly to Figure 6, the A_pk_ values in branches with direct contact with rods (616.7 ± 283.3 m/s^2^) were higher (Student’s *t*, *p* < 0.05) than branches without direct contact (268.1 ± 164.6 m/s^2^). The A_pk_ and A_RMS_ values showed a positive linear correlation (Pearson = 0.70, *p* < 0.05) in the tree canopy. In both cases, the direct contact of the rod represented an increment of 2.3 times the acceleration values reached in the branch. 

The canopy shaking system achieved a mean fruit detachment value of 69.1 ± 40.7%. However, this variable presented high variability within the cross-section of the tree hedge (Figure 6). As expected, this variable reached its highest values in the branches with direct contact with rods. The fruit detachment ratio was reduced from an average value of 84.7 ± 30.5% in branches with direct contact with rods to 25.1 ± 22.2% for branches without direct contact. The fruit detachment value of 100% was reached in all branches located between 2 and 2.5 m in height. A similar result was reported by Whitney et al. [6] who found that working with small canopies that were accessible to rods achieved a fruit detachment of 96%, while wider canopies achieved reduced values of 55%. Savary et al. [14] reached a fruit detachment value of 88% on the outside of the canopy, while on the inside this figure was reduced to 57%. However, in order to improve the harvesting efficiency, it is not only necessary for the rod to penetrate the canopy, but also for it to interact with the branch. Liu et al. [17], analysing fruit detachment according to the point of contact of the rod with the branch, determined that the operation was more effective when the rod impacted at 30% of the distance to the free end of the branch.

The results obtained from the cross-section of the tree canopy have shown a high variability, both in vibration and in fruit values. Reducing this variability and improving the mechanised harvesting process requires knowledge of the requirements to detach fruit from the tree. The measurements in the canopy showed a high linear correlation between variables, which indicated that fruit detachment prediction could not have a single solution. Analysis of the data focused on the discretisation of quantitative variables that could discriminate whether there was a fruit detachment with a value of 100% and reasonable success. Due to its simplicity and efficiency, logistic regression was used. A_RMS_ and time elapsed between an acceleration greater than 300 m/s^2^ (T_vib_300_) were significant variables to discriminate the events of 100% fruit detachment. The result was defined as a straight line that separates the conditions at which a fruit detachment of 100% was obtained with a precision measured as the area under the ROC curve of 0.95 in the validation set. Equation (1) shows the values obtained and Figure 7 is the graphical representation.
(1)$$ln\left(\frac{\mathrm{Prob}\;\mathrm{of}\;\mathrm{complete}\;\mathrm{fruit}\;\mathrm{detachment}\;}{1-\mathrm{Prob}\;\mathrm{of}\;\mathrm{complete}\;\mathrm{fruit}\;\mathrm{detachment}\;}\right)=7.13417-0.52754\;T_{vib_{300}}-0.17206\;A_{RMS}$$

Currently, canopy shaker systems for citrus harvesting can employ various types of vibration systems, with variations in the frequency or amplitude of movement, different machine ground speeds or rods with different designs, or mechanical properties. In all cases, the machine produces a forced vibration of the branches with the aim of detaching fruit and avoiding major damage to the tree or fruit. Monitoring of the forced vibration process of the tree canopy showed that it was possible to achieve a 100% fruit detachment ratio based on a combination of acceleration levels and vibration times in branches (Figure 7). Under the conditions of the field tests performed, we propose a combination of a vibration time of at least 5.8 s, after the interaction of the rod with the branch (T_vib_300_), together with an A_RMS_ value of 23.9 m/s^2^. With these harvest parameters, a complete process of fruit detachment was achieved in 88.9% of the branches tested. Although these values could be modified if another type of tree formation, citrus variety, or harvesting machine was considered, the fruit detachment process can be estimated by vibration time and branch acceleration. Both parameters are of great importance, not only for training trees to facilitate the rod penetration in the canopy but also for the design of new canopy shaker systems.

## 4. Conclusions

The monitoring of the vibration process in the tree during mechanised harvesting with the canopy shaker showed a great variability in results depending on different parts of the canopy. Branches that had direct contact with the machine rods showed a higher mean value of fruit detachment ratio (84.7%) than non-contact branches (25.1%). Vibration transmission from the external branches to the internal branches in the canopy was not effective to remove internal canopy fruit. During the harvesting process, values of 100% fruit detachment ratio could be achieved with a combination of harvest parameters in the branch. Achieving a complete process of fruit detachment is possible if both the tree canopy and machinery are mutually adjusted to facilitate the contact of the shaking system and the necessary vibration time.

## Figures and Tables

**Figure 1 sensors-19-01760-f001:**
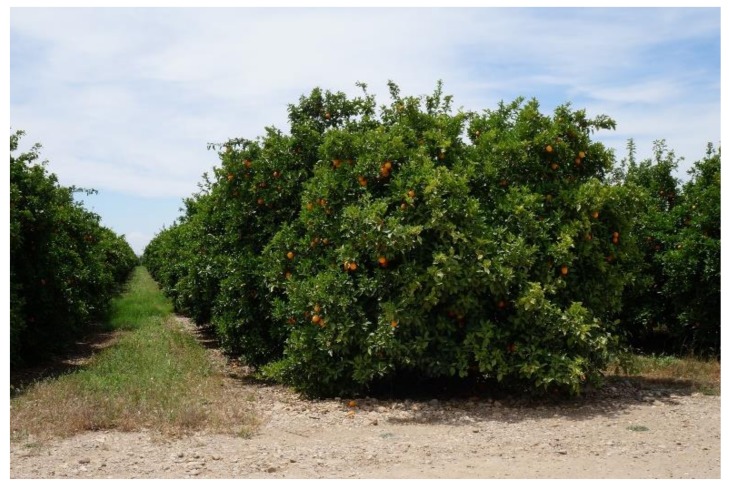
Example of a tree row trained in width hedge for mechanised harvesting.

**Figure 2 sensors-19-01760-f002:**
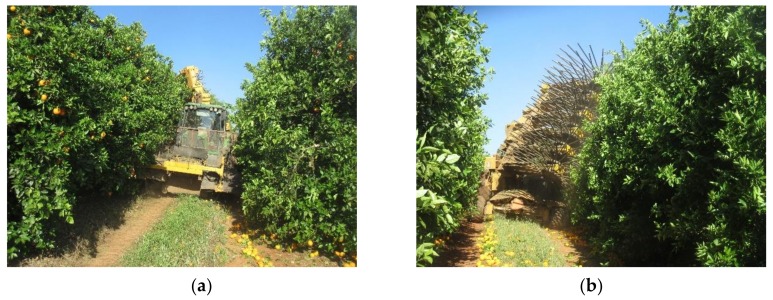
Lateral tractor-drawn continuous canopy shaker system (Oxbo, 3210) used in citrus harvesting tests. (**a**) front view before the harvesting process; (**b**) rear view after the harvesting process.

**Figure 3 sensors-19-01760-f003:**
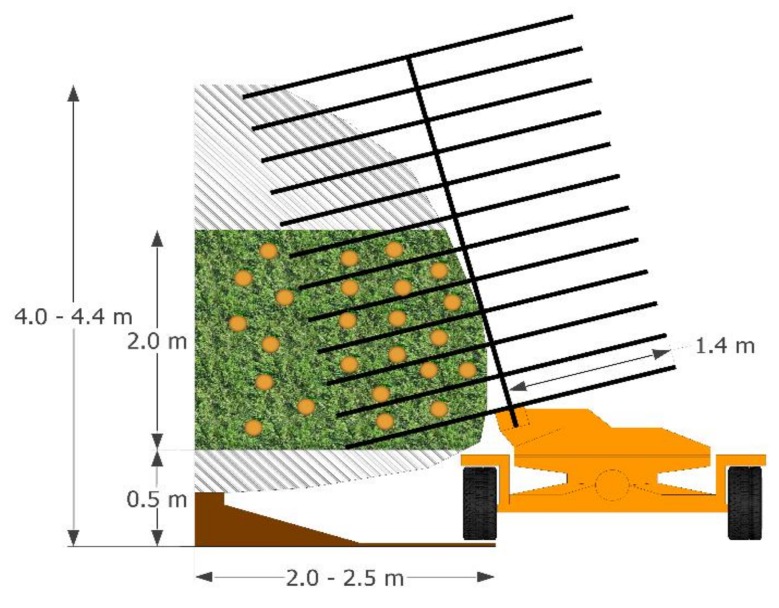
Cross-section of the tree hedge and canopy areas with and without direct contact with the canopy shaker system rods.

**Figure 4 sensors-19-01760-f004:**
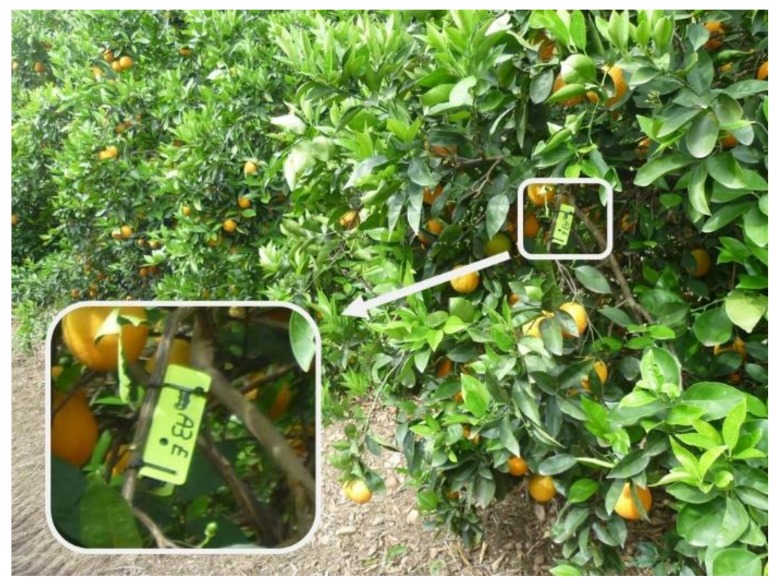
Location of the acceleration sensor on a fruit-bearing branch on the outermost part of the canopy.

**Figure 5 sensors-19-01760-f005:**
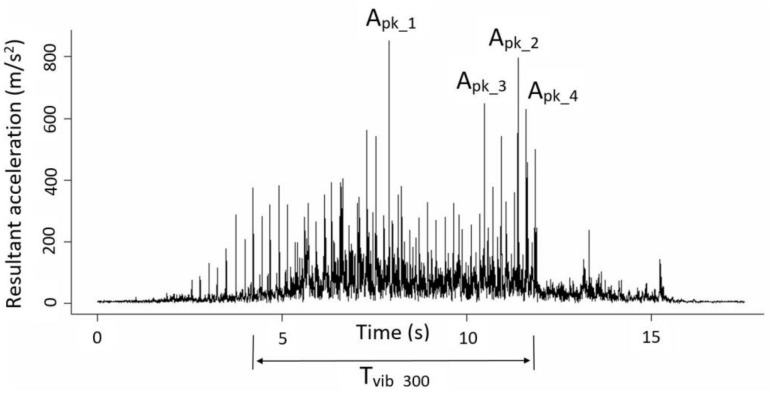
Example of the resultant acceleration (A_r_) in the time domain measured in a fruit-bearing branch. T_vib_300_—time elapsing between the first and the last event with an A_r_ value of 300 m/s^2^; A_pk_n_— n maximum peak value of A_r_.

**Figure 6 sensors-19-01760-f006:**
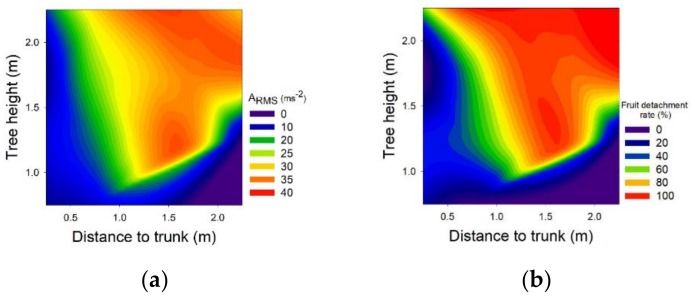
(**a**) Distribution of the A_RMS_ values (m/s^2^); and (**b**) fruit detachment rate (%) produced with a canopy shaker system and measured in fruit-bearing branches in the cross-section of the tree canopy.

**Figure 7 sensors-19-01760-f007:**
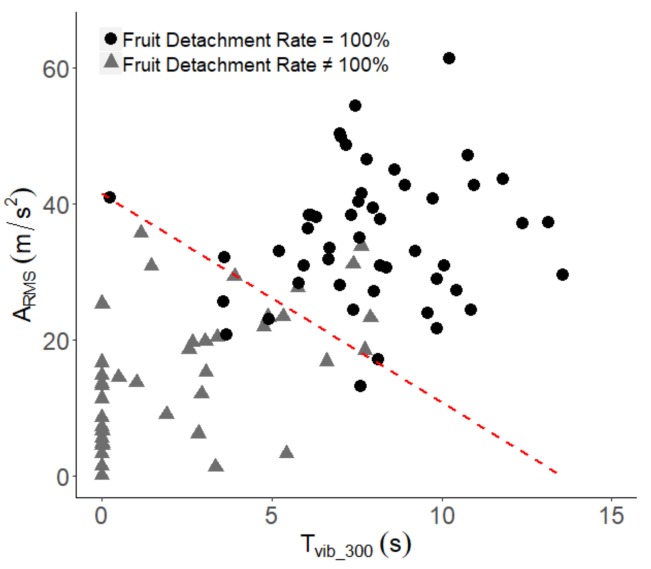
Distribution of the fruit detachment rate values according to A_RMS_ (m/s^2^) and T_vib_300_ (s).

**Table 1 sensors-19-01760-t001:** Characteristics of citrus orchards mechanically harvested with the canopy shaker system.

	Plot 1	Plot 2	Plot 3	Plot 4
Date planted	2006	2005	2007	2005
Plot area (ha)	54.7	38.0	33.1	57.3
Trees per ha	440	330	440	330
Tree distance (m)	7 × 3	7 × 4	7 × 3	7 × 4
Hedge height (m)	4.0	4.0	4.3	4.4
Hedge width (m)	3.9	4.1	4.5	4.6

**Table 2 sensors-19-01760-t002:** Vibration parameters measured on branches with and without direct contact with the rods during mechanical harvesting with the canopy shaker systems.

	Branches with Direct Contact with Rods	Branches without Direct Contact with Rods	Mean Value
Vibration time (s)	14.8 ± 2.8 ^a^	13.8 ± 2.9 ^a^	14.3 ± 2.8
Frequency (Hz)	4.1 ± 0.2 ^a^	4.0 ± 0.3 ^a^	4.1 ± 0.5
A_RMS_ (m/s^2^)	29.6 ± 10.2 ^a^	12.8 ± 6.4 ^b^	26.5 ± 13.6
Acceleration peak (m/s^2^)	616.7 ± 283.3 ^a^	268.1 ± 164.6 ^b^	495.1 ± 270.9
Fruit detachment ratio (%)	84.7 ± 30.5 ^a^	25.1 ± 22.2 ^b^	69.1 ± 40.7

Values shown are mean ± standard deviation, n = 90. The same superscript letters in the same row are not significantly different (Student’s *t*, *p* < 0.05; Wilcoxon–Mann–Whitney test, *p* < 0.05).

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
