# Peer review of "Vibration Monitoring of the Mechanical Harvesting of Citrus to Improve Fruit Detachment Efficiency"

_sensors, 2019, doi:10.3390/s19081760_

Round 1

Reviewer 1 Report

While this manuscript is an interesting read, it presents a case study. 

The only sensor applied is the one fixed to the three branch; time domain analysis is made. It is clear that if the excitation system is in direct contact with the branches, the peaks in time domain will be significantly higher then if there is no direct contact. The conclusions that the rate of detached oranges increases is obvious. There is nothing smart about this. 

I do not see scientific contribution to the community of the journal Sensors and cannot suggest publication. This manuscript should be submitted as a case study to a journal in the field of agriculture.

Author Response

Thank you for your review and comments. This work aims to provide a better understanding of the vibration process during the mechanical harvesting of citrus. Canopy shaker technology is widely spreading on intensive crops, especially when they are oriented to industrial transformation. In a few years, the hedge plantations configuration will be oriented to increase the mechanization, reducing the labor dependency. In fact, they are being used in important intensive woody crops, such as citrus and olives. The conditions of the study are representative of the current situation, both for the machinery and for the conditions and configuration of the orchards. From my point of view, it could be a representative case study. The key question is not to increase the vibration process to detach all fruit. We focus to understand where set the minimum combination of vibration parameters to make more efficiency the harvesting process. On the other hand, the machinery regulation and tree training are the output of the research.

Reviewer 2 Report

I have reviewed the manuscript sensors-473582 and I believe the authors have carried out an interesting analysis of the vibration parameters measured during mechanical harvesting of citrus. The data obtained during the vibration process offer accurate details of the measures, but I miss some aspects in the paper. I consider that there are some explanations that should be improved in order to a better understanding of the manuscript. My suggestions and comments are detailed below:

Mayor comments:

1.      As the title suggests, the goal of this research is to monitor the vibration produced in the mechanical harvesting of citrus to improve fruit detachment efficiency. Nevertheless, I consider the analysis presents few variations. For example:

a)      The mechanical harvest was carried out in June in all plots. I suppose that variations in the phenological stage (or even variations within the same phenological stage) can influence the detachment efficiency.

b)      The characteristics of the citrus plantations were very similar. As Table 1 shows, there are many similarities between the plantations of the four plots, with the exception of the year of plantation (maximum difference of 3 years) and hedge height (maximum difference of 0.7 m). This information is important as the distribution of fruit in the canopy depends on it.

c)      Why the authors did not contemplate the variability of branches and fruit within the tree canopy?.

d)      Why the authors did not consider the tree damage?.

2.       The results of the monitoring of the mechanical harvesting are very detailed, but the vibration parameters to improve the detachment efficiency are very vague. Although vibration parameters were set for the conditions of the field tests, there were no general parameters for other conditions. The two last sentences of the Conclusion section (L 321-324) are very general, but I think more concrete guides based on the plantation conditions should be define.

Minor comments:

L126. Revise the need for a superscript (4 to 5 m2).

Author Response

Thank you for revision and comments in order to improve the manuscript. Please, see in the attached doc file a point-by-point response to your revision. 

Reviewer 3 Report

I had the opportunity to review the manuscript "Vibration monitoring of the mechanical harvesting of citrus fruit to improve fruit detachment efficiency". The manuscript presents a scientifically relevant subject that is within the scope of Journal Sensors. This research will certainly contribute to the mechanical harvesting of juice oranges. However, some adjustments are necessary to make the manuscript clear to readers and can reach a large number of citations.

In the introduction, would it be possible to put more recent world production data for citrus, 2018 in the case?

I would like to see in the Introduction reports of other articles similar to this, which studied the canopy vibration process during mechanical harvesting of citrus .;

Please include a hypothesis of your study before the objective in the Introduction;

The scientific name Citrus sinensis should be highlighted in italics in the text;

The authors should mention the size of the experimental area sampled in the Methodology;

Include in the methodology the software used for statistical analysis;

I would like to see a boxplot with the evaluated variables;

It would also be possible to perform a principal component analysis to see which variables contributed to the greater variability of the data. These analyzes may allow authors to expand their discussion of their results.

Author Response

(The authors gave the same response as above.)

Round 2

Reviewer 1 Report

I am sorry, but I do not see significant improvements in the re-submitted version. This research is more a case study to me. I cannot suggest publication.

Reviewer 2 Report

Thanks for the efforts to improve the manuscript.

Reviewer 3 Report

The authors made the proposed corrections and improved the manuscript, which is in a position to be accepted.